# Long Term Memory Network for Combinatorial Optimization Problems

## Abstract

This paper introduces a framework for solving combinatorial optimization problems by learning from input-output examples of optimization problems. We introduce a new memory augmented neural model in which the memory is not re-settable (i.e the information stored in the memory after processing an input example is kept for the next seen examples). We used deep reinforcement learning to train a memory controller agent to store useful memories. Our model was able to outperform hand-crafted solver on Binary Linear Programming (Binary LP). The proposed model is tested on different Binary LP instances with large number of variables (up to 1000 variables) and constrains (up to 700 constrains).

## 1 Introduction

An intelligent agent with a long-term memory processes raw data (as images, speech and natural language sentences) and then transfer these data streams into knowledge. The knowledge stored in the long-term memory can be used later in inference either by retrieving segments of memory during re-calling, matching stored concepts with new raw data (e.g. image classification tasks) or solving more complex mathematical problems that require memorizing either the method of solving a problem or simple steps during solving. For example, the addition of long-digit numbers requires memorizing both the addition algorithm and the carries produced from the addition operations (Zaremba et al., 2016).

In neural models, the weights connecting the layers are considered long term memories encoding the algorithm that transform inputs to outputs. Other neural models as recurrent neural networks (RNNs) introduce a short-term memory encoded as hidden states of previous inputs (Lipton et al., 2015; Hochreiter & Schmidhuber, 1997).

In memory augmented neural networks (MANNs), a controller writes memories projected from its hidden state to a memory bank (usually in the form of a matrix), the controller then reads from the memory using some addressing mechanisms and generates a read vector which will be fed to the controller in the next time step (Graves et al., 2014). The memory will contain information about each of the input sequence tokens and the controller enriches its memory capacity by using the read vector form the previous time step.

Unfortunately, In MANNs the memory is not a long-term memory and is re-settable when new examples are processed, making it unable to capture general knowledge about the inputs domain. In context of natural language processing, one will need general knowledge to answer open-ended questions that do not rely on temporal information only but also on general knowledge from previous input streams. In long-digits multiplication, it will be easier to store some intermediate multiplication steps as digit by digit multiplications and use them later when solving other instances than doing the entire multiplication digit by digit each time from scratch.

Neural networks have a large capacity of memorizing, a long-term persistent memory will even increase the network capacity to memorize but will decrease the need for learning coarse features of the inputs that requires more depth.

Storing features of the inputs will create shortcut paths for the network to learn the correct targets. Such a network will no longer need to depend on depth to learn good features of the inputs but instead will depend on stored memory features. In other words a long-term memory can provide

intermediate answers to the network. Unlike regular MANNs and RNNs, a long-term memory can provide shortcut connections to both inputs features and previous time steps inputs.

Consider when the memory contains the output of previous examples, the network would cheat from the memory to provide answers. Training such a network will focus on two stages: (1) Learning to find similarities between memory vectors and current input data, (2) learning to transform memory vectors into meaningful representations for producing the final output.

The No Free Lunch Theorem of optimization (Wolpert & Macready, 1997) states that: any two algorithms are equivalent when their performance is averaged across all possible problems, this means that an algorithm that solve certain classes of problems efficiently will be incompetent in other problems. In the setting of combinatorial optimization, there is no algorithm able to do better than a random strategy in expectation. The only way an algorithm outperforms another is to be specialized to a certain class of optimization problems (Andrychowicz et al., 2016). Learning optimization algorithms from scratch using pairs of input-output examples is a way to outperform other algorithms on certain classes. It is further interesting to investigate the ability of learned models to generate better solutions than hand crafted solvers.

The focus of this paper is on designing neural models to solve Binary Linear Programming (or 0-1 Integer Programming) which is a special case of Integer Linear Programming problems where all decision variables are binary. The 0-1 integer programming is one of Krap's 21 NP-complete problems introduced in Karp (1972). The goal of Binary LP is to optimize a linear function under certain constrains. It is proved by Cadoli (2001) that Binary LP expresses the complexity class NP (i.e any problem in the complexity class NP can be modeled as Binary LP instance).

The standard form of a Binary LP problem is:

$$\begin{aligned} \text{maximize} \quad & c^T x \\ \text{subject to} \quad & Ax \leq b \\ & x \geq 0 \\ & x \in \{0, 1\} \end{aligned}$$

where $c$ and $b$ are vectors and $A$ is a matrix.

We propose a general framework for long-term memory neural models that uses reinforcement learning to store memories from a neural network. A long-term memory is not resettable and may or may not store hidden states from individual time steps. Instead a long term memory stores information that is considered to be useful for solving similar instances. The controller that decides to write memories follows a policy function that properly constructs the memory contents. We train and test this framework on synthetic data set of Binary LP instances. We analyze the model capability of generalization to more complex instances beyond the training data set.

## 2    LITERATURE REVIEW

A related line of work is learning to learn and meta-learning, Lake et al. (2016) argued that it is an important building block in artificial intelligence. Younger et al. (2001) and Andrychowicz et al. (2016) used recurrent neural networks to act as a gradient descent procedure to train other networks.

The application of neural networks to combinatorial optimization has a long distinguished history. Most common approach is using Hopfield networks (Hopfield & Tank, 1985) for solving Travelling Salesman Problem (TSP). Other neural approaches applied to Travelling Salesman Problem include Elastic nets (Durbin, 1987) and Self-organized maps (Fort, 1988).

Pointer networks (Vinyals et al., 2015) an architecture similar to sequence-to-sequence neural network (Sutskever et al., 2014) to solve combinatorial problems as TSP. Pointer networks solve class of problems where the number of target classes in each step of the output depends on the input length which is variable. Recently, Bello et al. (2016) introduced a pointer network for solving combinatorial problems which is optimized using policy gradient methods. This pointer network was applied to TSP and Knapsack.

Our work is different from this recent approaches, we applied our model to a general mathematical formulation of combinatorial optimization problems, instead of learning solutions from problem

dependent data. We introduce a memory neural network where the coupled memory is persistent and is not resettable. The model is able to learn and store inputs features as neural memories and utilizes them in solving Binary LP instances. The model can learn from minimal amount of data and yet generate better solutions than handcrafted solver.

## 3 LONG-TERM MEMORY NETWORK FOR BINARY LP

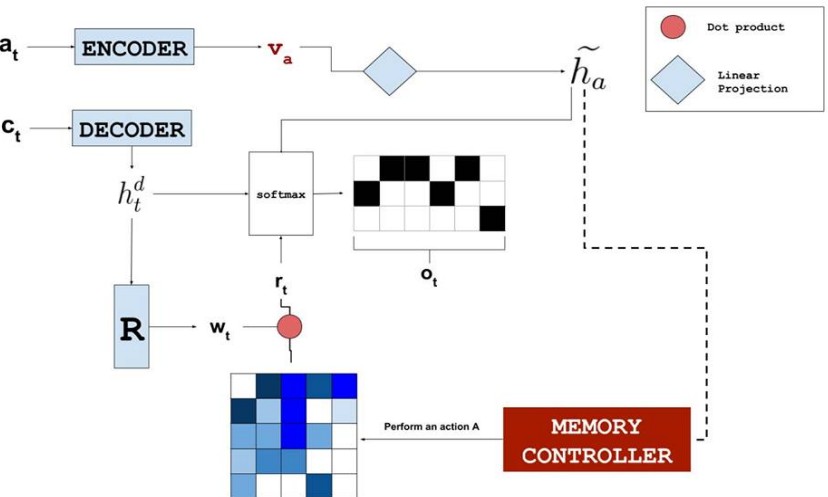

Figure 1: The operation of an LTMN processing one Binary LP instance. First the encoder receives a sequence of constrains coefficients and produces a memory vector via a linear projection of encoder final hidden state. The memory controller receives this linear projection and decides whether to store it in memory $M$ or delete the current slot it points to. A control signal (sequence of costs) is then passed to the decoder to produce un-normalized weights $w_t$ over the memory locations. A dot product operation between $w_t$ and $M$ Finally the output $o_t$ at each time step is generated using the memory vector (or encoder final state), the decoder hidden state and the read vector.

A sequence-to-sequence approach to solve Binary LP requires formulating the entire problem inputs (costs vector $c^T$, bounds vector $b$ and constrains matrix $A$) as one sequence and mapping it to an output sequence. A naive linear programming solver constructs the set of feasible solutions from the constrains matrix $A$, we denote this set as $S_F$, then it iterates over $S_F$ using the cost function to find the optimal solution. For a Binary LP instance, the number of possible feasible solutions is $2^N$ where $N$ is the number of variables.

The model we describe works in in a similar way to the naive solver, an encoder encodes the entire constrains matrix into one fixed vector $v^a$, this vector is a continuous representation of the set $S_F$. The vector $v^a$ is then passed as the initial state of a decoder along with each cost $c_t$ that is used as a control signal to read from memory. Figure 1 describes the operation of an LTMN for a single Binary LP instance.

A general Long-Term Memory Network (LTMN) consists of three basic modules: a core network, a memory bank in the form of a matrix and a memory controller. The core network has three main functions: providing memory fragments that are useful information from input signals to be stored in the memory, providing a reading mechanism to read from the memory, and finally producing an output. The memory bank is a matrix $R \times C$, were each row is addressed by a pointer $PTR$ which is a value in range $[0, R-1]$. The pointer is provided to the memory controller to know the last accessed memory slot. Clearing the memory contents is not allowed by the memory controller.

A memory controller is another module that uses reinforcement learning as a hard attention mechanism to store memory fragments from the core network as in Zaremba & Sutskever (2015). Upon processing of each example, the controller basically chooses either to store or discard the memory fragments. Writing to the memory in LTMN is totally a discrete process, the final computational

graph will not be fully differentiable. Thus it is essential to pass the memories to the output layer for backpropagation to work.

The final output depends on both memory contents representation (a read vector) and inputs embedding, this is similar to end-to-end memory networks (Sukhbaatar et al., 2015) that is used in question answering tasks. In end-to-end memory networks the final answer depends on both the output memory representation and question embedding.

For non-sequential data the core network can be a regular feed forward neural network and the memories can be the hidden vectors from each layer, while for sequential data the core network can be a recurrent neural network and memories can be its hidden states at each time step. The attention mechanism (Bahdanau et al., 2014) used over to attend to memory locations will depend either on a control signal or previous output.

## 3.1 ENCODER

Let the sequence $A = \{a_{11}, a_{12}, ..., a_{1N}, SEPARATOR, b_i, SEPARATOR, ..., a_{MN}\}$ be the sequence describing the constrains set, $a_{ij}$ represents the coefficient of the $j^{th}$ variable in the $i^{th}$ constrain. The $SEPARATOR$ token separates between the left and right hand sides of the constrains inequality, the same token is used to separate between constrains in the sequence. Finally $b_i$ is the bound for the $i^{th}$ constrain. The encoder module encodes the entire sequence $A$ in to a fixed vector. The encoder then produces a linear projection of this vector to be stored in memory. Equations (1) and (2) describe the constrains encoder operation.

$$v^a = C(a_t, \Theta_c) \tag{1}$$

The linear projection of $v^a$ is:

$$\widetilde{h_a} = W_{hv}v^a + b_h \tag{2}$$

where $W_{hv}$ and $b_h$ are weight matrix and bias vector respectively.

## 3.2 DECODER

The input to the decoder is a sequence of costs $C = \{c_1, ...c_N\}$. The decoder uses the fixed vector $v^a$ as initial hidden state. At each time step the decoder reads a cost $c_t$ of the costs vector $C$ and produces a hidden state vector:

$$h_t^d = D(c_t, \Theta_d) \tag{3}$$

The reader uses the decoder hidden state $h_t^d$ to produce weights that are used to address the memory:

$$w_t = R(h_t^d; \Theta_r) \tag{4}$$

A simple dot product operation between the memory and the weight vector produces a read vector:

$$r_t = w_t \cdot M^{(i)} \tag{5}$$

where $M^{(i)}$ is the memory produced by the memory controller after processing instance $i$ in the data set.

The linear projection $\widetilde{h_a}$, the cost embedding $w_t$ and the read vector $r_t$ are used to generate a final output:

$$o_t = O(\widetilde{h_a}, w_t, r_t; \Theta_o) \tag{6}$$

The final output layer is a soft-max layer, and the whole model can be thought of estimating the probability distribution $P(o_t | \widetilde{h_a}, w_t, r_t)$. Each cost is embedded via a linear layer first, then the embedding is passed to recurrent layers. The reader is implemented as a recurrent layer, in this sense the weights produced will depend on both the cost at time step $t$ (the control signal) and the weight from the previous time step $w_{t-1}$.

The reader produces un-normalized weights over memory slots $N$ in range [-1,1]. When the weight for a slot $i$ is 0 this means that the memory slot does not contribute at all to the read vector, when it is 1 the whole memory slot is used. The interesting case is when the weight is $-1$ which means the inverse of the current memory slot contents, when the weight is between between 0 and 1 the

information from the memory slot is preserved to a certain extent, instead a $-1$ weight transforms the entire slot into a new representation. Hopefully, the reader weights learned by backpropgation through the whole network will not only act as attention weights over memory slots but also transform the memory contents in to useful representations.

### 3.3 MEMORY CONTROLLER

A typical memory controller is provided a memory fragment at a discrete time $t$ and makes a decision whether to store or discard this memory. Learning to store useful memories is descirbed as an reinforcement learning (RL) framework. An agent senses the environment **s**, takes an action **a** from an action space **A** using a policy function $\pi$ and receives an immediate reward **r**. The total future reward from point t is expressed by:

$$R_t = \sum_{l=0}^{\infty} \gamma^l r_{t+l} \tag{7}$$

where $\gamma$ is the discount factor.

#### 3.3.1 ENVIRONMENT AND ACTIONS

The environment is the memory bank at discrete time step $t$ along with the current pointer $PTR_t$ and a window of previous memories $W_t$ that the controller has been subjected to. A window is similar to a short term memory with limited size $\tau$ and the controller chooses to store one of the contents of that memory. In all our experiments, the controller stores the last memory vector in the window, in this sense the window gives insights to the controller about what memories have been produced by the core network.

The basic actions a controller may perform are storing the current memory we refer to this as $a_{STR}$, and not to store any thing at all we refer to this as $a_{NO-OP}$. We include three more actions: delete the current slot $a_{DELETE}$, increment the pointer then store $a_{STRINC}$, and decrement the pointer then store $a_{STRDEC}$.

#### 3.3.2 REWARD FUNCTION

An action taken by the controller is evaluated through a reward function. One can evaluate the controller the same way we evaluate the core network, through common metrics as final loss or accuracy, for each example the model solves correctly the controller gets a reward. In this way, the rewards received will only account for how better the whole model gets on solving tasks. Instead we want the controller to store useful memories that will be used in the next processed examples. The reward function is a simple question: **Did the current memory result in a better response than empty memory ?**. The reward function is:

$$r_t = \begin{cases} 1 & if\ eval(A, C, M_t) = 1\ \&\ eval(A, C, M_{Zero}) = 0 \\ 0 & otherwise \end{cases} \tag{8}$$

where $M_{Zero}$ is the memory with zero entries, $M_t$ is the memory produced by memory controller and $eval$ is an evaluation function for the core network. When the entries of a read vector are all zeros the output will depend only on information coming from the control signal (costs $c_t$ in case of Binary LP). The memory controller receives a reward only when the non-empty memory results in solving an example correctly and empty memory results in incorrect response. An optimal training procedure will make the core network depends only on non-empty memory to minimize a loss function, while an optimal training procedure for the memory controller will result in useful memories only.

## 4 MEMORY CONTROLLER TRAINING

At the heart of the memory controller is an RL agent which interacts with the environment. The environment is basically the memories received per each example the LTMN solves. For each instance the LTMN processes, the RL agent receives a state $s_t$ and selects an action $a_t$ following a policy $\pi(a_t|s_t)$ and receives a reward $r_t$ and the next state $s_{t+1}$ (Sutton & Barto, 1998).

To train the memory controller agent, we used deep Q-learning algorithm with experience replay (Mnih et al., 2013) as described in algorithm 1. A neural network is used to estimate the action-value function $Q(s, a)$ (Tsitsiklis & Van Roy, 1996; Sallans & Hinton, 2004; Riedmiller, 2005). We choose to implement the action-value neural network as a stacked LSTM which receives the memory contents $M_t$, one slot each time step, followed by window contents $W_t$ and the pointer $PTR$.

We draw the similarity between using deep Q-learning algorithm for playing Atari games and using it for storing long-term memories. During each episode the core network is given a sequence of instances form a training dataset $X_{controller}$ and the memory agent should take proper actions on memory to construct useful memory entries that helps solving theses instances. One can consider the agent is learning to construct the memory as constructing a puzzle from pieces, where the pieces come from a bucket. There will be no final state for the agent to reach, because of the fact that one can not know exactly how the memory should be constructed. In the setting of Atari games the agent will reach the final state (game over) quickly in the first few epochs as the agent will still be struggling to learn an optimal action-value function. In the case of the memory agent, we have to determine an episode length $T$ that both simulates the agent failure in the earlier training phase and the agent reaching a final state. The final state of the agent will be reached when the LTMN processes the last example in an episode. Thus, there will be an infinite number of final states. We can keep the episode length constant so the agent have a limited time to succeed in accumulating rewards.

Both the reader and the memory controller agent should be trained jointly together. This procedure is critical since both the reader and the memory controller depend on each other, the reader learns to generate proper weights over memory slots that depend on a control signal and the memory controller learns how to construct memories that are useful for the reader. For each action the memory agent perform on the memory, one should train the core network. We sample a batch of instances from training data $X_{train}$, run the memory agent using this batch for one session (memory is not cleared between instances) and perform the backward pass on the core network.

A typical learning algorithm for neural networks as stochastic gradient descent, updates the network parameters using data batches. The number of iterations (one forward pass followed by one backward pass) is equal to $\lceil N/m \rceil$ where $N$ is the size of dataset and $m$ is the batch size. The number of iterations can get large as the size of the dataset increases.

To effectively train the LTMN core network, the episode length should be as the same as the number of iterations. In our experiments, we keep the episode length $T$ as small as possible and then increase it after $K$ epochs. To avoid similar episode lengths and simulate various solving sessions (where the memory controller agent has to store memories), each episode $T$ is chosen randomly between $[k, \lfloor N/m \rfloor]$, where $k$ is the minimum length of an episode.

## 5 EXPERIMENTS

### 5.1 DATA GENERATION

We generate two separate data sets: one for training the core network and the other for training the memory controller. We generate 14k Binary LP instances as the core network training set, 3k instances as the memory controller training set and 3k as a validation/test set. We use mixed curriculum learning strategy as suggested in Zaremba et al. (2016). For each instance, we generate two random numbers $n$ and $m$, where $n$ is the number of variables and $m$ is the number of constrains for the instance. The maximum number of variables in an instance is $N$ and maximum number of constrains is $M$. In our experiments, $n$ is chosen uniformly between $[3, N]$ and $m$ is between $[1, M]$. For our training dataset $N$ is 10 and $M$ is 5.

All the coefficients of the objective function and the constrains matrix $A$ is generated between [-99,99]. To ensure that the constrains matrix is sparse we generate a random number $SL$ called sparsity level, for each constrain in the problem we ensure that at most $1/3$ of the coefficients are zeros. To generate supervised targets for the problems we used a python interface to the COIN-OR branch and cut solver (Lougee-Heimer, 2003) called PuLP (Mitchell & Dunning, 2011). We ensure that all the generated problems have a feasible solution. We use denote the COIN-OR branch and cut solver as the baseline solver and compare the solver results with the LTMN results.

---

**Algorithm 1** LTMN Training Algorithm

---

1:  Initialize replay memory $D$ to capacity $N$
2:  Initialize the stacked LSTM $Q$ with random weights
3:  **for** episode = 1,M **do**
4:      Select randomly an episode length T
5:      **for** t = 1,T **do**
6:          Select randomly an example $x_i$ from $X_{controller}$
7:          Solve $x_i$ using Encoder to get memory vector $\widetilde{h_{ai}}$
8:          Store $\widetilde{h_{ai}}$ in to window $W_t$
9:          With probability $\epsilon$ select a random action $a_t$
10:          otherwise select $a_t = max_a Q^*(s_t, a; \theta)$
11:          Execute action $a_t$ on memory $M_t$ observe the next memory state $M_{t+1}$ and reward $r_t$
12:          Set $s_t = M_t, W_t, PTR_t$
13:          Set $s_{t+1} = M_{t+1}, W_t, PTR_{t+1}$
14:          Store transition $(s_t, a_t, r_t, s_{t+1})$ in $D$
15:          Sample random minibatch of transitions $(s_t, a_t, r_t, s_{t+1})$ from $D$
16:          Perform a backward pass on $Q$
17:          Sample random minibatch of training data $x_b$ from $X_{train}$
18:          Perform a forward pass of $Q$ on $x_b$ and Set $M_{x_b}$
19:          Perform a backward pass on LTMN using $x_b$ and $M_{x_b}$
20:      **end for**
21: **end for**

---

## 5.2 EXPERIMENTAL DETAILS

We implement both the encoder and decoder as a recurrent neural network with gated recurrent units (GRU). We use three GRU layers each with 256 units followed by a linear layer with 128 units to project the memory vector of the constrains sequence. As suggested by Bello et al. (2016), it is harder to learn from input-output examples of optimization problems due to subtle features that model cannot figure out. We suggest a technique to learn more features by connecting the $1, 2, ..., L - 1$ layers to layer $L$ instead of connecting only the previous layer $L - 1$. In this way an encoder can learn more features using combined features from the previous layers.

The decoder is implemented as two stacked GRU layers, the reader is implemented as one GRU layer. The costs are first embedded using a linear layer of 64 units. The rest of the decoder has 256 units. All the weights are initialized uniformly in [-0.08,0,08]. We set the dropout ratio to 0.2 in all GRU layers. We constrain the norm of the gradients to be no greater than 5 (gradient clipping). We set the learning rate initially to 0.001 and drop it by factor 0.5 after each 10 epochs.

The memory controller agent is implemented as a stacked LSTM of 2 layers each with 200 units. We used batch normalization over the inputs to normalize the memory contents and window contents to zero-mean and unit variance. We used Adam (Kingma & Ba, 2014) optimizer to train the agent. The window size $\tau$ is 5 in all our experiments. We used generalization loss as an early stopping criteria for the core network as suggested by Prechelt (2012). We then allow the memory controller agent to train until a reasonable average total reward per episode is reached.

We compare the performance of LTMN to the baseline solver and the sequence-to-sequence approach. We trained a network of GRU units with comparable number of parameters (seq-to-seq has 1.3M parameters and LTMN has 1.9M parameters). The seq-to-seq network receives the input as one long sequence.

We train the LTMN core network and seq-to-seq model using RMSprop algorithm (Tieleman & Hinton, 2012) with cross entropy as a loss function.

Table 1: Average costs at different sampling temperatures $T$

| average costs of the solver = 24.76 | | |
|---|---|---|
| Sampling Temperature $T$ | Average Costs | No of Instances with a better solution |
| 0.6 | 51.345 | 575 |
| 1.0 | 42.938 | 546 |
| 1.2 | 43.056 | 565 |
| 1.5 | 36.646 | 544 |
| 2.2 | 29.417 | 503 |

Table 2: Average costs for the untrained model

| average costs of the solver = 24.76 | |
|---|---|
| Sampling Temperature $T$ | Average Costs |
| 1.0 | 1.421 |
| 1.2 | 3.841 |
| 1.5 | 0.517 |
| 2.2 | -2.574 |

## 5.3 RESULTS

We define a metric to test the quality of solutions produced by the model, we define average costs over $N$ Binary LP instances:

$$Average\,Costs = \frac{\sum_{i=0}^{N} Cost(i)}{N} \tag{9}$$

All the instances in training data and testing data are maximization problems, so the higher the average costs the better. We sample only feasible solutions using the output probability distribution with temperatures $[0.6, 1.0, 1.2, 1.5, 2.2]$.

First we evaluate the LTMN model against the baseline solver, we generate a test set of 1000 instances where all instances have 10 variables and the number of constrains is between 1 and 5. Table 1 shows at different sampling temperatures the average costs and the number of instances where the model outperformed the solver and found a better solution. The LTMN model outperformed the baseline solver by large margin (average costs are nearly $51.7\%$ higher).

To validate that the sampling technique is effective, we used an initial untrained model to sample solutions. Table 2 shows that the untrained model performed poorly on the same test set, and hence the random sampling is not generating solutions by chance and the trained model learned to generate solutions effectively.

We also evaluate the LTMN against the seq-to-seq GRU model which fails to generate any feasible solutions. While the number of instances is small (only 14K instances), other similar models as pointer networks (Vinyals et al., 2015) used a training data of 1M instances for Travelling Salesman Problem, however our model did not require large training dataset.

We test the generalization ability of LTMN to longer sequences beyond those in training set. Figure 2 shows the results on test sets of Binary LP instances (each test set contains 1000 instances) where the number of variables is incremented while the maximum number of constrains is 5. LTMN outperformed the baseline solver by large margin even when the number of variables is larger than 100, the LTMN was still able to generate better solutions.

We test LTMN generalization on very large instances. Table 3 shows that LTMN outperformed the baseline solver even for very large instances with 1000 variables and 700 constrains.

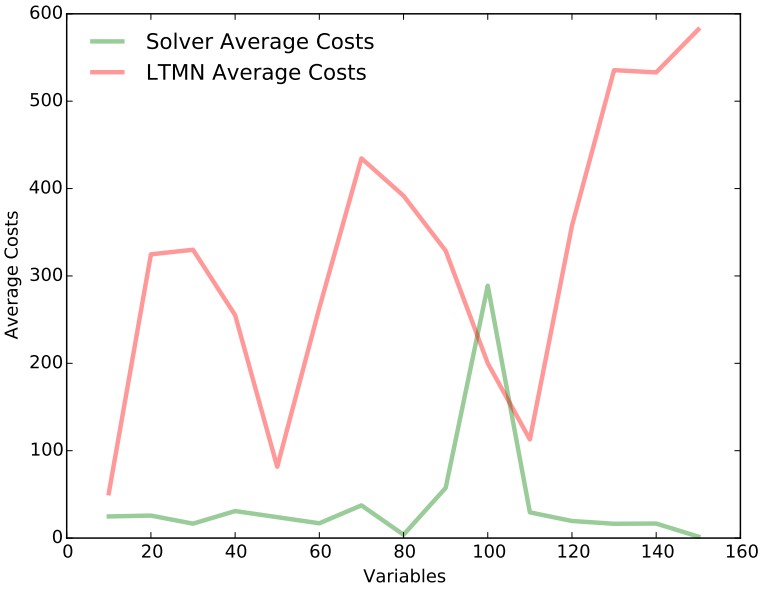

(a) Average costs of LTMN vs baseline solver

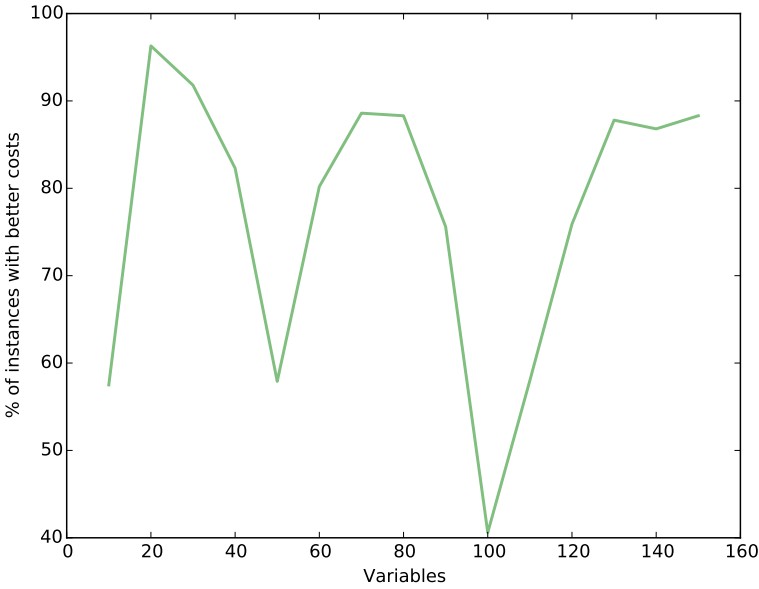

(b) Percentage of instances with better costs (higher than baseline solver)

Figure 2: Evaluation of LTMN on test sets of Binary LP instances with number of constrains in range $[1, 5]$ and number of variables up to 150

## 5.4 MEMORY TESTS

To understand the effectiveness of using an augmented long-term memory network for solving Binary LP instances, we conduct tests to prove that long-term memory improves model results. We calculate the average costs for the same test set (10 variables and constrains between 1 and 5) but

Table 3: Objective values generated by baseline solver and object values calculated using the sampled variables assignments from LTMN output probability distribution at different sampling temperatures for large Binary LP instances.

| | | | LTMN objective values at different T | | | | |
|---|---|---|---|---|---|---|---|
| N | M | Solver | T=0.6 | T=1.0 | T=1.2 | T=1.5 | T=2.2 |
| 100 | 50 | 441 | **387** | 599 | 462 | 420 | 286 |
| 300 | 100 | 1026 | 1513 | **467** | **942** | **1107** | 785 |
| 500 | 700 | 1175 | **1866** | **1334** | 2062 | 2263 | 1754 |
| 1000 | 350 | 870 | **4189** | **3694** | **4247** | **4341** | 3309 |

Table 4: Average costs (memory is reset between examples)

| average costs of the solver = 24.76 | |
|---|---|
| Sampling Temperature $T$ | Average Costs |
| 0.6 | 46.896 |
| 1.0 | 42.526 |
| 1.2 | 39.727 |
| 1.5 | 35.621 |
| 2.2 | 28.063 |

we reset the memory each time a new example is processed. Table 4 shows that the average costs is slightly dropped when the memory is reset between examples.

We conduct a per-example test where we identify whether memory helped in generating good solutions or not. Given two problems identified by their cost vectors and constrains matrices as $[c_1, A_1]$ and $[c_2, A_2]$. Let $A_1 \neq A_2$ but both of them construct the same set of feasible solutions $S_F$. The encoder produces two memory vectors $\widetilde{h}_{A_1}$ and $\widetilde{h}_{A_2}$ for constrains $A_1$ and $A_2$ respectively. To ensure that both $\widetilde{h}_{A_1}$ and $\widetilde{h}_{A_2}$ represent the same set of feasible solutions $S_F$, we measure the cosine similarity between these two memory vectors such that $CosSim(\widetilde{h}_{A_1}, \widetilde{h}_{A_2}) \geq S$, where $S$ is set to 0.8. We then enforce the controller to store $\widetilde{h}_{A_1}$, and generate a solution for $[c_2, A_2]$

We define three metrics:

**Memory Faults**: number of examples where non-empty memory results in worse solution than empty memory.

**Memory Trues**: number of examples where non-empty memory results in better solution than empty memory.

**Memory Equals**: number of examples where both non-empty memory and empty memory result in the same solution.

We generate 10K instances (with 10 variables and constrains in range of [1,5]), we compare the feasible solutions of each two consecutive instances and calculate the cosine similarity between the two memory vectors. We found 181 instances with the same feasible solutions. We record the solutions of these instances using both empty memory and non-empty memory containing $\widetilde{h}_{A_1}$.

Table 5 shows high memory trues where non-empty memory helped generating better solutions than empty memory. The table also shows high percentage of memory equals, for these 28 instances the non-empty memory did not help much in generating a better solution, in fact both memory faults and memory equals are similar metrics and can be thought of as the number of instances where non-empty memory failed to generate a better solution. A good memory controller agent should maximize the memory trues and minimize both the memory faults and equals at the same time.

We conclude that a long-term memory is quite effective in generating better solutions and the memory controller learns effectively how to store input features useful for longer interaction with a model.

Table 5: Memory faults, memory trues and memory equals

| Metric | Value | Percentage |
|---|---|---|
| Memory faults | 66 | ≈36.4% |
| Memory trues | 87 | ≈48.0% |
| Memory equals | 28 | ≈15.4% |

## 6 CONCLUSION

This paper introduced a long term memory coupled with a neural network, that is able to memorize useful input features to solve similar instances. We applied LTMN model to solve Binary LP instances. The LTMN was able to learn from supervised targets provided by a handcrafted solver, and generate better solutions than the solver. The LTMN model was able to generalize to more complex instances beyond those in the training set.

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
