# OpenReview forum: "Long Term Memory Network for Combinatorial Optimization Problems"
_ICLR.cc/2018/Conference — Reject_

### Official Review · AnonReviewer3 · 2017-11-27
**Very hard to judge**

**Rating:** 4
**Confidence:** 1

**Review:**

Learning to solve combinatorial optimization problems using recurrent networks is a very interesting research topic. However, I had a very hard time understanding the paper. It certainly doesn’t help that I’m not familiar with the architectures the model is based on, nor with state-of-the-art integer programming solvers.

The architecture was described but not really motivated. The authors chose to study only random instances which are known to be bad representatives of real-world problmes, instead of picking a standard benchmark problem. Furthermore, the insights on how the network is actually solving the problems and how the proposed components contribute to the solution are minimal, if any.

The experimental issues (especially regarding the baseline) raised by the anonymous comments below were rather troubling; it’s a pity they were left unanswered.

Hopefully other expert reviewers will be able to provide constructive feedback.

---

### Official Review · AnonReviewer2 · 2017-11-27

**Rating:** 4
**Confidence:** 2

**Review:**

# Summary
This paper proposes a neural network framework for solving binary linear programs (Binary LP). The idea is to present a sequence of input-output examples to the network and train the network to remember input-output examples to solve a new example (binary LP). In order to store such information, the paper proposes an external memory with non-differentiable reading/writing operations. This network is trained through supervised learning for the output and reinforcement learning for discrete operations. The results show that the proposed network outperforms the baseline (handcrafted) solver and the seq-to-seq network baseline.

[Pros]
- The idea of approximating a binary linear program solver using neural network is new.

[Cons]
- The paper is not clearly written (e.g., problem statement, notations, architecture description). So, it is hard to understand the core idea of this paper.
- The proposed method and problem setting are not well-justified.
- The results are not very convincing.

# Novelty and Significance
- The problem considered in this paper is new, but it is unclear why the problem should be formulated in such a way. To my understanding, the network is given a set of input (problem) and output (solution) pairs and should predict the solution given a new problem. I do not see why this should be formulated as a "sequential" decision problem. Instead, we can just give access to all input/output examples (in a non-sequential way) and allow the network to predict the solution given the new input like Q&A tasks. This does not require any "memory" because all necessary information is available to the network.
- The proposed method seems to require a set of input/output examples even during evaluation (if my understanding is correct), which has limited practical applications.

# Quality
- The proposed reward function for training the memory controller sounds a bit arbitrary. The entire problem is a supervised learning problem, and the memory controller is just a non-differentiable decision within the neural network. In this case, the reward function is usually defined as the sum of log-likelihood of the future predictions (see [Kelvin Xu et al.] for training hard-attention) because this matches the supervised learning objective. It would be good to justify (empirically) the proposed reward function.
- The results are not fully-convincing. If my understanding is correct, the LTMN is trained to predict the baseline solver's output. But, the LTMN significantly outperforms the baseline solver even in the training set. Can you explain why this is possible?

# Clarity
- The problem statement and model description are not described well.
1) Is the network given a sequence of program/solution input? If yes, is it given during evaluation as well?
2) Many notations are not formally defined. What is the output (o_t) of the network? Is it the optimal solution (x_t)?
3) There is no mathematical definition of memory addressing mechanism used in this paper.
- The overall objective function is missing.

[Reference]
- Kelvin Xu et al., Show, Attend and Tell: Neural Image Caption Generation with Visual Attention

---

### Official Review · AnonReviewer1 · 2017-11-27
**Interesting approach but flawed experiments**

**Rating:** 3
**Confidence:** 4

**Review:**

This paper proposes using long term memory to solve combinatorial optimization problems with binary variables. The authors do not exhibit much knowledge of combinatorial optimization literature (as has been pointed out by other readers) and ignore a lot of previous work by the combinatorial optimization community. In particular, evaluating on random instances is not a good measure of performance,  as has already been pointed out. The other issue is with the baseline solver, which also seems to be broken since their solution quality seems extremely poor. In light of these issues, I recommend reject.

---

### Public Comment · (anonymous) · 2017-10-27
**Missing related work**

If you're going to compare against combinatorial optimization problems, you should cite work on combinatorial optimization - this is an active area of research and the google paper was rejected last year for ignoring previous work and overstating contributions.

Is the COIN-OR package anywhere close to SOTA? I'd expect the industrial solvers like CPLEX and Gurobi to be far better, and they have free academic licenses available. You're showing an improvement in 60% of cases with only ten variables in Figure 2, but there's only 2^10 = 1024 possible variable values, meaning it's possible to brute force search through all the solutions and achieve the optimal value. The fact that your baseline solver doesn't do that suggests it's not a very strong baseline.

---

### Public Comment · (anonymous) · 2017-11-03
**Interesting approach but very poor methodology**

The authors propose to use a long-term memory network to solve combinatorial optimization problems, namely binary linear programs.

Using a long-term memory is definitely interesting, at it may capture a deeper understanding of the combinatorial problems at hand. This knowledge may in turn help improve the resolution process.

That being said, this works suffers from seemingly clear lack of knowledge of optimization-related literature and practices.

* As was mentioned in a previous comment, there is no mention of existing literature on combinatorial optimization and integer linear optimization. The claim that "the application of neural networks to combinatorial optimization has a long and distinguished history" is only supported by references to works that are 30 years appart (mid 80s and mid 2010s).

* The authors state that (sec. 3, 1st paragraph) "a naive linear solver constructs the set of feasible solutions, [...] then iterates over [it] using the cost function to find the optimal solution". This wrongly suggests that linear solvers go through explicit enumeration, which is not the case. Cutting planes and branch-and-bound techniques should be mentioned, or at least referred to (any textbook on linear programming would have a chapter on this).

* The experimental procedure (section 5) goes against most good practices from the OR community:
    - A set of randomly-generated instance is NOT representative of any real-life problem (see MIPlib for a well-known and broadly-used benchmark), and such instances are likely to be infeasible. This latter concern is not mentionned.
    - In the generated dataset, "at most 33% of coefficients are non-zeros". Practical instances are much more sparse (see MIPlib instances)
    - The baseline appears to be extremely poor, as was pointed out by a previous comment. 10 variables means at most 1024 solutions. Any solver that does not achieve optimality of such instances should not be considered as a baseline.
    - The performance of the algorithms is evaluated using the average cost as a metric (with no mention of variance in the results). This is not a good metric for it is too sensitive to extreme values. Performance profiles (see Dolan and Moré 2002) are a more comprehensive tool for benchmarking optimization algorithms.


All in all, the lack of relevant litterature and poor methodology raise serious concerns over the contribution of the proposed approach.

---

> ### Public Comment · (anonymous) · 2017-11-13
> **Strongly agree**
>
> Looking forward to a response from the authors.
>
> Another signal that the baseline is broken - with 80 and 150 variables, the average cost is zero, which means that for all of the 1000 problems at these two sizes, their baseline returned the trivial all zeros solution.
>
> The baseline is clearly broken (in addition to the many troubling concerns already pointed out in other comments).

---

### Decision · Program_Chairs · 2018-01-29
**ICLR 2018 Conference Acceptance Decision**

**Decision:**

Reject

**Comment:**

The authors use a memory-augmented neural architecture to learn solve combinatorial optimization problems.  The reviewers consider the approach worth studying, but find the authors' experimental protocol and baselines flawed.